# A Role for *COX20* in Tolerance to Oxidative Stress and Programmed Cell Death in *Saccharomyces cerevisiae*

**DOI:** 10.3390/microorganisms7110575

**Published:** 2019-11-18

**Authors:** Ethiraju Keerthiraju, Chenyu Du, Gregory Tucker, Darren Greetham

**Affiliations:** 1School of Biosciences, University of Nottingham, Sutton Bonington Campus, Loughborough LE12 5RD, UK; stxke1@nottingham.ac.uk (E.K.); gregory.tucker@nottingham.ac.uk (G.T.); 2School of Applied Sciences, University of Huddersfield, Queensgate, Huddersfield HD1 3DH, UK; c.du@hud.ac.uk

**Keywords:** yeast, oxidative stress, respiratory growth, programmed cell death, COX20

## Abstract

Industrial production of bioethanol from lignocellulosic materials (LCM′s) is reliant on a microorganism being tolerant to the stresses inherent to fermentation. Previous work has highlighted the importance of a cytochrome oxidase chaperone gene (*COX20*) in improving yeast tolerance to acetic acid, a common inhibitory compound produced during pre-treatment of LCM’s. The presence of acetic acid has been shown to induce oxidative stress and programmed cell death, so the role of *COX20* in oxidative stress was determined. Analysis using flow cytometry revealed that COX20 expression was associated with reduced levels of reactive oxygen species (ROS) in hydrogen peroxide and metal-induced stress, and there was a reduction in apoptotic and necrotic cells when compared with a strain without COX20. Results on the functionality of COX20 have revealed that overexpression of COX20 induced respiratory growth in Δimp1 and Δcox18, two genes whose presence is essential for yeast respiratory growth. COX20 also has a role in protecting the yeast cell against programmed cell death.

## 1. Introduction

During fermentation, yeast is exposed to a range of cellular stresses such as accumulation of toxic end products such as ethanol or butanol, a shift into more acidic pH, anaerobic growth conditions and nutrient limitation [1]. Oxygen during the fermentation process is required for the synthesis of unsaturated fatty acid and sterols; however, oxygen can be rapidly depleted [2,3]. Reactive oxygen species (ROS), such as superoxide anion (O_2_^−^), hydrogen peroxide (H_2_O_2_), and hydroxyl radicals (OH) are generated as by-products of cellular metabolism [4]. During fermentation, the presence of ROS can disrupt a diverse array of biological processes [5] and has been shown to damage a variety of cellular components, including DNA, proteins, and unsaturated lipids [6]. ROS has been shown to have a direct role in cellular aging [7]; the cellular lifespan has been related to the anti-oxidant potential of the cell and the number of re-pitches into fermentation [8,9].

Yeasts’ response to the presence of ROS is diverse, with an enzymatic response, such as the upregulation of thioredoxins, glutaredoxins, catalases, superoxide dismutase, and a non-enzymatic cellular response (such as glutathione becoming oxidized and then being recycled) [10,11,12]. COX20 is a mitochondrial inner membrane protein, whose known function is to chaperone COX2, a subunit of cytochrome c oxidase in the yeasts mitochondrial matrix [13]. COX20 is an absolute requirement for the efficient export of COX2; this interaction also requires COX18 [13]. COX20 has also been shown to stabilize unassembled COX2 against degradation by an i-AAA protease [14]. Previous work has shown that *COX20* is upregulated in the presence of acetic acid, and its overexpression protects the cell against hydrogen peroxide-induced oxidative stress [15]. Programmed cell death in yeast can be induced by the addition of acetic acid, whose presence can induce a cytochrome c cascade from the mitochondria. 

In this paper, populations of yeast cells in the presence of a variety of oxidative stress-inducing conditions were looked at when exposed to hydrogen peroxide or metals in cells overexpressing COX20 or being COX20 deficient.

## 2. Material and Methods

### 2.1. Strains, Media, and Growth Conditions

Yeast strains employed in this work were derived from *Saccharomyces cerevisiae* BY4741 and have been described previously [15]. These consisted of a Δ*cox20* deleted BY4741 yeast strain that had been transformed with either an empty vector control (Δ*cox20* (pCM173)) or a vector expressing *COX20* (Δ*cox20*(pCM173:*COX20*)). Yeasts were grown in synthetic drop-out medium (SD) without tryptophan (Sigma-Aldrich, St. Louis, MI, USA). Additionally, *IMP1* and *COX18* were deleted from these strains using a pAG34 hygromycin selectable marker (Euroscarf, Oberursel, Germany) and primers, COX18 knockout forward 5′-GGATCCATGTTAAAGAGGTTAGCCAA-3′, COX18 knockout reverse 5′-CCCGGGTCATCGTTGGTAAGGATAAA-3′ IMP1 knockout forward 5′-AAGCTTATGACGGTTGGTACACTTCC-3′ and IMP1 knockout reverse 5′-CCCGGGTCAGTTGCTCTTAGCCTGCAC-3′, respectively. COX18 and IMP1 knockouts were grown in SD-trp, in the presence of 50 µg/mL hygromycin. Strains were grown in SD-trp media containing 30 g/L^−1^ glycerol rather than glucose for respiratory assays.

### 2.2. Spot Plate Assays

Cryopreserved yeast cells were propagated overnight in 5 mL SD-trp as above. The cultures were centrifuged at 17,000× *g* for 4 min at 4 °C. The pellet was washed with 5 mL sterile distilled water and resuspended in 100 μL of sterile water. Optical density (OD600) was measured and adjusted to an OD600 of 1.0, corresponding to 1.0 × 10^7^ cells/mL. Cells containing a vector were then serially diluted ten-fold, and 5 μL aliquots from each 10-fold dilution was spotted onto SD-trp agar containing hydrogen peroxide. BY4741-*trp1* was grown on SD media. All plates were incubated at 30 °C for 48 h. Observations were made at 24 and 48 h, respectively. The assay was performed in triplicate.

### 2.3. Determination of GSH and GSSG Levels 

Cells were grown in SD-trp medium to an OD600 OD of 0.5, and treated with oxidant (1 mM hydrogen peroxide or 0.5 mM metal sulphates, i.e., FeSO_4_) or no oxidant for control cells for 15 min, the cells were then harvested via centrifugation (5 min; 1500× *g*). For the estimation of intracellular glutathione, cells were washed twice with ice-cold phosphate-buffered saline (pH 7.4) resuspended in ice-cold 1.3% (*wt*/*vol*) 5-sulfosalicylic acid and 8 mM HCl, and broken using a 3000 mp ultrasonic homogenizer (Biologics, Inc, Cary, NC, US) using 10 s amplitude pulses for 1 min with the sample on ice at all times. The resulting cell-free extract was clarified by centrifugation (10 min, 13,000× *g* at 4 °C), and glutathione levels were measured in the resulting supernatant using a microplate method. Briefly, the reduction of 1.7 mM 5,5′-dithio-bis (2-nitrobenzoic acid (DTNB) in the presence of GSSG-reductase (3.3 U/mL) was measured for 5 min at 412 nm, the experiment was initiated by the addition of 0.9 mM β-NADPH. For quantification of oxidized glutathione, samples (including GSSG standards) were pretreated with 5% (*vol*/*vol*) 2-vinylpyridine for 1 h at room temperature before analysis. Glutathione levels are expressed as nanomoles per OD600, where OD600 of 1 corresponded to a cell density of 1.0 × 10^7^ cells mL^−1^.

### 2.4. ROS Detection Kit

The total ROS detection kit (Enzo Life Sciences Ltd., Exeter, UK) was used for ROS detection. The assay monitors real-time ROS production in live cells using flow cytometry and included Oxidative Stress Detection Reagent (Green). Upon staining, the fluorescent product generated can be visualized using a flow cytometer equipped with a blue (488 nm) laser. This non-fluorescent, cell-permeable total ROS detection dye reacts directly with a wide range of reactive species, such as hydrogen peroxide, peroxynitrite, and hydroxyl radicals, yielding a green fluorescent product indicative of the cellular presence of different ROS or reactive nitrile species (RNS) moieties. However, this assay does not detect superoxide, chlorine, or bromine species.

### 2.5. Cell Preparation for Flow Cytometry

Cryopreserved cells were inoculated into 5 mL SD-trp medium and incubated for 48 h at 30 °C. These were then transferred to 50 mL of SD-trp and grown until cells were in the exponential phase of growth (OD 0.5) ~6 h in a 250 mL conical flask with shaking, at 30 °C. To ensure that the cells were in the log phase before starting the experiment, they were cultured to a density not exceeding 1 × 10^6^ cells/mL^−1^.

Cells were centrifuged at 400× *g* for 5 min and the supernatant removed. Cells were resuspended in 1× Wash Buffer at a concentration of 5 × 10^5^ cells/mL^−1^ and 0.5 mL added per sample into flow tubes. Cells were simultaneously treated with an experimental test agent (or control) and exposed to the ROS Detection Solution. ROS Detection was carried out using a Beckman Coulter FC500 (Beckman Coulter Ltd., Wycombe, UK). Cells under increased oxidative stress are characterized by bright green fluorescence in the presence of the Oxidative Stress Detection Reagent. 

Apoptotic cells were detected using Annexin V labeled with fluorescein (FITC) (λabs/λem = 492/514 nm) in green by binding to phosphatidylserine. Ethidium homodimer III (EtD-III) is a highly positively charged nucleic acid probe, which is impermeable to live cells or apoptotic cells, but stains necrotic cells with red fluorescence (λabs/λem = 528/617 nm). Cells were prepared as above for the detection of ROS. Cells that stain green and red are progressing from apoptotic cell populations into necrotic. 

### 2.6. Statistics

ANOVA was determined using XLSTAT, a statistical software package designed for use with Excel (https://www.xlstat.com/en/). The significance in the Figures was denoted as * for *p* 0.05%, ** for *p* 0.01%, and *** *p* <0.001, respectively.

## 3. Results

### 3.1. Overexpression of COX20 Protects Yeast Cells from Oxidative Stress 

The stress response of BY4741-*trp1*, Δ*cox20*(pCM173), and Δ*cox20*(pCM173:*COX20*) strains was determined under control, 500 μM pyocyanin (PCN) (general ROS inducer), and 1 mM hydrogen peroxide (Figure 1). Previously published data have shown significant overexpression of *COX20* using a pCM173 plasmid when compared with very low or no COX20 expression observed in a BY4741-*trp1* control strain [15].

It was determined that there was no difference in yeast populations under control conditions (*p* = 0.8472) (Figure 1A–C). Assays with 0.5 M pyocyanin (0.5 M was chosen as this concentration has been shown to induce oxidative stress in *S. cerevisiae* BY4741 [16]), revealed that 22.92 ± 0.25% of BY4741 cells were redox stressed, deletion of *cox20* increased the presence of redox stressed yeast cells to 37.5 ± 0.35%; however, overexpression of *COX20* reduced the number of redox stressed cells to 16.09 ± 0.48% (Figure 1D–F). Assays with 1 mM hydrogen peroxide, a concentration which has previously been shown to induce oxidative stress in yeast, revealed that 44.36 ± 1.02% BY4Y4741-*trp1* cells were redox stressed (Figure 1G), which compared with Δ*cox20* null mutants (70.18 ± 2.15%) (Figure 1H) and cells overexpressing *COX20* (29.21%), respectively (Figure 1I).

The spread of the distribution is usually expressed as the standard deviation (SD), however, in flow cytometry, the coefficient of variation (CV) is preferred because it is dimensionless and, on a linear scale, does not depend on where in the histogram the data is recorded [17]. The CV analysis revealed that the CV for Δ*cox20* (pCM173) was higher than the CV observed for the Δ*cox20* (pCM173:*COX20*) strain in response to 1 mM hydrogen peroxide and PCN, indicating that under these conditions there were higher levels of ROS present in this strain (Figure 2). In contrast, the CV profiles for both the strains under control conditions were not significantly different (*p* = 0.6201). 

Overexpression of *COX20* protected the yeast cell to hydrogen peroxide induced oxidative stress. Oxidative stress can also be induced in a yeast cell by the presence of metals [18], so the importance of *COX20* when yeast cells are exposed to metals was determined. It was determined that overexpression of *COX20* protected the yeast cell against a range of copper concentrations (0.1–0.5 mM) (Figure 3A), cobalt (0.1–0.5 mM) (Figure 3B), iron (0.1–0.5 mM) (Figure 3C), manganese (0.25–1.5 mM) (Figure 3D), cadmium (0.1–0.5 mM) (Figure 3E), zinc (0.1–0.5 mM) (Figure 3F). There was no difference in the presence of silver (0.1–2 mM) between cell lines used in this paper (data not shown).

### 3.2. COX20 Overexpression Protects the Cell from Glutathione Becoming Oxidized

The presence of hydrogen peroxide and metals has been shown to cause toxicity through depletion of GSH in a yeast cell, so GSH and oxidized GSH (GSSG) levels in BY4741-*trp1* or in a *cox20* null mutant overexpressing either *COX20* or an empty vector were determined. Under control conditions, there was a decrease in GSH and the concurrent increase in GSSG in yeast cells containing an empty vector when compared with BY4741-*trp1* cells, indicating a more oxidized environment in yeast without *COX20* (Figure 4A–C) (indicated by red asterisks *p* > 0.001). Overexpression of *COX20* increased GSH and concurrently decreased GSSG levels when compared with BY4741-*trp1* (Figure 4A–C). Presence of hydrogen peroxide and some metals (with the exception of zinc, cadmium, and silver) reduced GSH content and increased GSSG in Δ*cox*20(pCM173) when compared with BY4741-*trp1* cells (Figure 4A–C) (indicated by black asterisks *p* > 0.001); however, overexpressing *COX20* recovered this phenotype and indeed made the yeast cell more reduced than that observed in the BY4741-*trp1* cells (Figure 4A–C). This effect of reducing GSH and increasing GSSG made a Δ*cox*20 (pCM173) cell more oxidized when compared with BY4741-*trp1* cells, and overexpression made the cells more reduced than the control cell line (Figure 4C).

### 3.3. Overexpression of COX20 Reduced the Presence of Apoptotic, Late Apoptotic, and Necrotic Yeast Cells

Oxidative stress has been shown to induce programmed cell death [19], so we determined if *COX20* has any role in protecting the cell against programmed cell death. The determination of apoptotic and necrotic cells under control conditions revealed no differences between strains overexpressing *COX20* or empty vector controls (Figure 5A–C). A Δ*cox20* null mutant expressing an empty vector contained a higher percentage of apoptotic, late apoptotic, and necrotic cells in the presence of hydrogen peroxide (*p* < 0.001) (Figure 5A–C) and a higher percentage of apoptotic cells in the presence of cobalt and copper (*p* < 0.001) (Figure 5A) when compared with BY4741-*trp1*. Overexpression of *COX20* reduced the presence of apoptotic cells in assays with hydrogen peroxide, cobalt, copper, manganese iron and cadmium when compared with BY4741-*trp1* (all *p* <0.001) (Figure 5A). There was no effect of overexpressing *COX20* in regards response to zinc or silver (Figure 5A). Overexpression of COX20 reduced the presence of late apoptotic cells in the presence of hydrogen peroxide, cobalt, copper, manganese, iron, cadmium, and silver when compared with BY4741-*trp1* cells (*p* <0.001 for all except silver (*p* <0.05)). Overexpression of *COX20* reduced the presence of necrotic cells in the presence of hydrogen peroxide, cobalt, copper, manganese, iron, and cadmium (Figure 5C). In assays with hydrogen peroxide, BY4741-*trp1* had significantly reduced the occurrence of necrotic cells when compared with BY4741-*trp1* Δ*cox20* (pCM173) (*p* > 0.005); however, overexpression of *COX20* still reduced necrotic cell populations significantly when compared with BY4741-*trp1* (*p* > 0.005) (Figure 5C). There was no difference in apoptotic, early necrotic, or necrotic cells in the presence of zinc. 

There was a significant difference in the presence of late apoptotic cells in a Δ*cox*20(pCM173) when compared with BY4741-*trp1* or Δ*cox20* (pCM173:*COX20*) cells (*p* < 0.001); however, presence of silver had no difference in the presence of oxidatively stressed cells in these cell lines (data not shown). 

### 3.4. Investigating the Role of COX20 in the Mitochondrial Response to Oxidative Stress

Cox20p, as stated previously, is a chaperone for the folding of Cox2p in the mitochondria, unfortunately, *COX2* is mitochondrially encoded and is therefore difficult to make a null mutant of this gene. Cox20p is required for the N-terminal processing of Cox2 by the mitochondrial inner-membrane proteins (Imp1p, Imp2p, and Som1p) with Cox18p being responsible for moving the tail of Cox2p into the intermembrane space [20]. An excellent model of how Cox2p is processed has been previously constructed and was used as a guide in this research [21]. In particular, we wished to discern Cox20p′s role in protecting the yeast cell was as a result of it binding to Cox2p or how the Cox20p; Cox2p complex interacted with Cox18p. The model showed that proteins such as Imp1 were important to how Cox2p and Cox20p bound together. We looked at tolerance to hydrogen peroxide induced oxidative stress in yeast cells overexpressing *COX20* in strains lacking *IMP1* or *COX18*. Spot tests showed that disrupting *IMP1* in a strain overexpressing *COX20* returned the strain back to peroxide sensitivity, whereas knocking out *COX18* had no effect on tolerance (Figure 6A). 

Assays revealed that deleting *COX20* significantly increased the number of apoptotic and necrotic cells when compared with BY4741 cells (Figure 6B). Overexpression of *COX20* reduced the presence of apoptotic or necrotic cells significantly (*p* < 0.001) (Figure 6B). The deletion of *IMP1* had no effect on the presence of necrotic or apoptotic cells as a single knockout when compared with a BY4741 strain; however, deletion of IMP1 in a Δ*cox20* (pCM173:*COX20*) increased the presence of apoptotic and necrotic cells (*p* < 0.001) (Figure 5B). Assays with *COX18* revealed no effect on programmed cell death when compared with a Δ*cox20* strain or a strain overexpressing *COX20,* indicating that this enzyme had no role in response to programmed cell death with Cox20p. Deleting *IMP1* or *COX18* had no effect on the levels of apoptotic or necrotic cells when compared with control strains (BY4741-*trp1*), indicating that the lack of IMP1 or COX18 did not promote apoptosis or necrosis away from the absence of COX20 (Figure 6B). Determining the redox status of the cells in the presence of hydrogen peroxide revealed that a Δ*cox20*Δ*imp1* (pCM173:*COX20*) cell line was more oxidized when compared with either Δ*cox20* (pCM173:*COX20*) or Δ*cox20*Δ*cox18* (pCM173:*COX20*) (Figure 6C). 

Yeast cells lacking *IMP1* or *COX18* have been shown to be incapable of growing in respiratory conditions, so we assayed for the effect of overexpressing *COX20* when growing on glycerol. The results revealed that a Δ*cox20* null mutant was capable of growing on glycerol; however, it was significantly slower than the BY4741-*trp1* (Figure 5D) (*p* = 0.031). Overexpressing *COX20* in a Δ*cox20* null mutant recovered the ability to grow on glycerol (*p* = 0.041); however, this strain was still slower than BY4741-*trp1*. The assays with Δ*imp1* or Δ*cox18* confirmed that the deletion of these genes prevents utilization of glycerol, and therefore, respiratory growth (Figure 6D); however, expression of *COX20* in these strains resulted in growth on glycerol being observed (Figure 5D). There was no difference for any of these strains when grown on 2% glucose (data not shown).

## 4. Discussion

Previous work has shown that a *COX20* deleted yeast strain was extremely sensitive to acetic acid stress and that transformation of this strain with a vector designed to express *COX20* increased tolerance above that observed for the parental strain [15]. *COX20* expression in wild type has been shown to increase in response to acetic acid [15]. *COX20* encodes for a chaperone that facilitates proteolytic processing of the mitochondrial gene product COX2 in terms of its assembly into the mitochondrial inner membrane cytochrome c oxidase complex [13]. Yeast strains that lack glutathione or are altered in their GSH redox state are sensitive to oxidative stress caused by reactive oxygen species ROS [22]. *S. cerevisiae* undergoes programmed cell death (PCD) in response to the presence of acetic acid during fermentation [23]; in this response, cytochrome c is released early from intact mitochondria initiating a mediated apoptotic cascade mechanism [24,25]. At a later stage in the process, oxidative phosphorylation is blocked, and the cytochrome c is degraded [25]. Oxidative stress during fermentation has been shown to damages a yeast′s ability to tolerate high temperatures [26] and cause membrane lipid peroxidation [27]. Additionally, ROS accumulation and oxidative damage have been observed in enological yeast strains when fermenting high-sugar containing media [28]. 

One suggested mechanism for acetic acid toxicity in this cascade is that it induces oxidative stress. Given the known function of *COX20*, this could also be implicated in resistance to oxidative stress. Deletion of *COX20* had no discernible effect on the yeast under control (non-stressed) conditions; however, in the presence of hydrogen peroxide, a Δ*cox20* deleted strain was sensitive when compared with a strain expressing *COX20*. Results revealed that the expression of *COX20* improved tolerance to the presence of hydrogen peroxide. Tolerance to oxidative stress was observed in assays looking at metabolite activity and growth. Furthermore, ROS levels in response to hydrogen peroxide appeared to be reduced in *COX20* expressing cells suggesting a role for *COX20* in the cell’s response to oxidative stress. The study was extended to look at these yeast strains′ response to the presence of metals, with the exception of zinc and silver. There was a reduction in the populations of oxidatively stressed and apoptotic and necrotic cells. The toxic effects of transition metals such as Fe, Cu, Co, Mn have the effect of generating hydroxyl radicals which can cause damage to cellular components, including the mitochondria [29]. In contrast, Zn has antioxidant effects, which are antagonistic to the redox activities of Fe and Cu, and thus, the effect is not associated with oxidative stress in the mitochondria [30]. 

The mitochondrial cytochrome oxidase complex of yeast is complex and consists of at least 12 subunits, respectively. The core components of the complex are the subunits COX1, COX2, and COX3 which form the catalytic core of the enzyme. COX2 requires COX20 for chaperoning; without COX20, the proteolytic maturation of COX2 cannot proceed [13]. The COX2 precursor is matured by the IMP1 protease, which is dependent on COX20 binding to Cox2 [13]. *COX2* is a difficult gene to work with as it is one of the few mitochondrially encoded genes, looking at the genes involved in the processing of COX2 allowed us to investigate where COX20 might respond to oxidative stress. Assays revealed that the deletion of *COX18* had no effect on yeast cells overexpressing *COX20* ability to tolerate oxidative stress, whereas the deletion of *IMP1* returned yeast back to sensitivity. This result indicated that COX20′s role in responding to oxidative stress is post-binding to COX2. COX18 translocates the COX2 C-tail to the inner mitochondrial space in a mechanism which is currently unknown, this activity leads to a functioning cytochrome c oxidase complex [31], and however, results here indicate that the complex may respond to the presence of free-radicals independent of activities of this enzyme and involving the action of IMP1.

Clearly, COX20 has further roles within the mitochondria; a structured approach looking at the genes encoding proteins involved in the process would be required before a full understanding of the mechanism by which COX20 protects cells against oxidative stress. Results here indicate that the action of COX20 may be related to its binding to COX2, making COX20 important in the absence of the COX18. Further assays looking at COX2 in these knockout strains and cytochrome C oxidase activity are required before a complete understanding of the role of COX20 in the mitochondria. This study has shown the importance of *COX20* in how yeast responds to redox-induced stress and a further study into the expression levels of COX20 during fermentation and discovering yeast strains with inherently higher *COX20* expression may identify yeast which can ferment at higher temperatures and at higher sugar concentrations. Wine fermentations in countries such as Spain are currently struggling due to the higher summer temperatures, and wine-makers are constantly searching for new yeast to increase wine production and product diversity [32]. This study suggests yeasts that express COX20 will be more tolerant of these stresses and may become a desirable physiological trait in the future. 

## Figures and Tables

**Figure 1 microorganisms-07-00575-f001:**
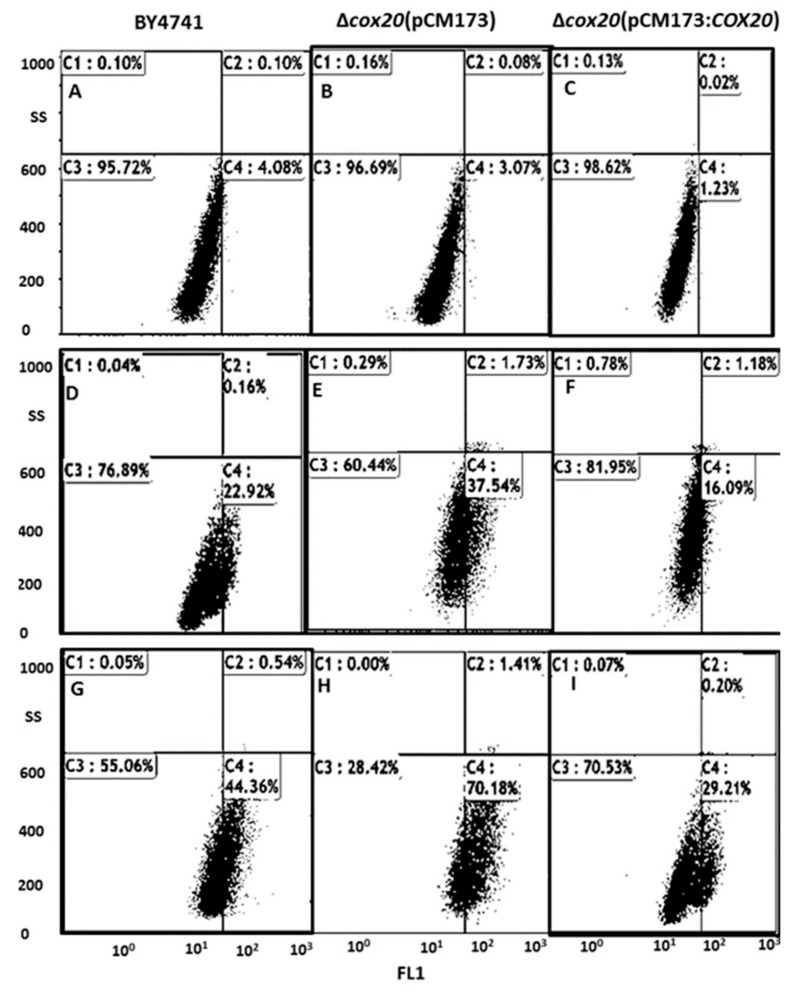
Dot plot diagrams of yeast populations using a flow cytometer with quadrants for (**A**) BY4741-*trp1* control, (**B**) Δ*cox20* (pCM173) control, (**C**) Δ*cox20*(pCM173:*COX20*) control, (**D**) BY4741-*trp1*, (**E**) Δ*cox20* (pCM173), (**F**) Δ*cox20* (pCM173:*COX20*) in the presence of 0.5 mM pyocyanin, (**G**) BY4741-*trp1*, (**H**) Δcox20 (pCM173:COX20), (**I**) Δ*cox20* (pCM173:*COX20*) in the presence of 1 mM hydrogen peroxide.

**Figure 2 microorganisms-07-00575-f002:**
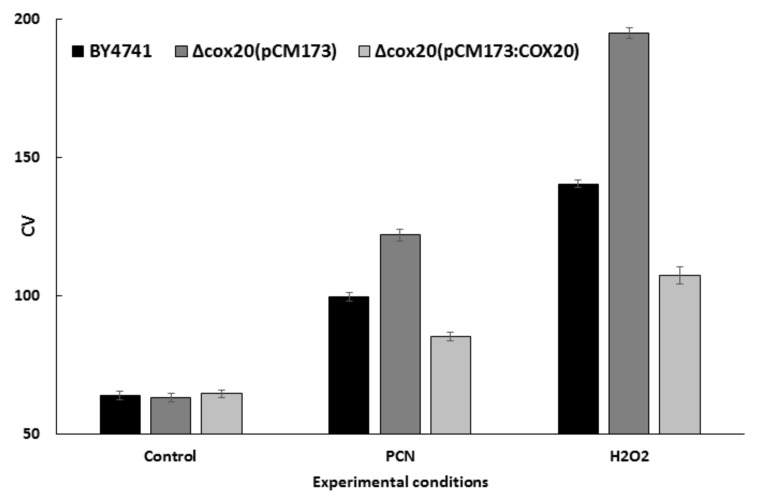
Coefficient of variation analysis of data derived from flow cytometry assays.

**Figure 3 microorganisms-07-00575-f003:**
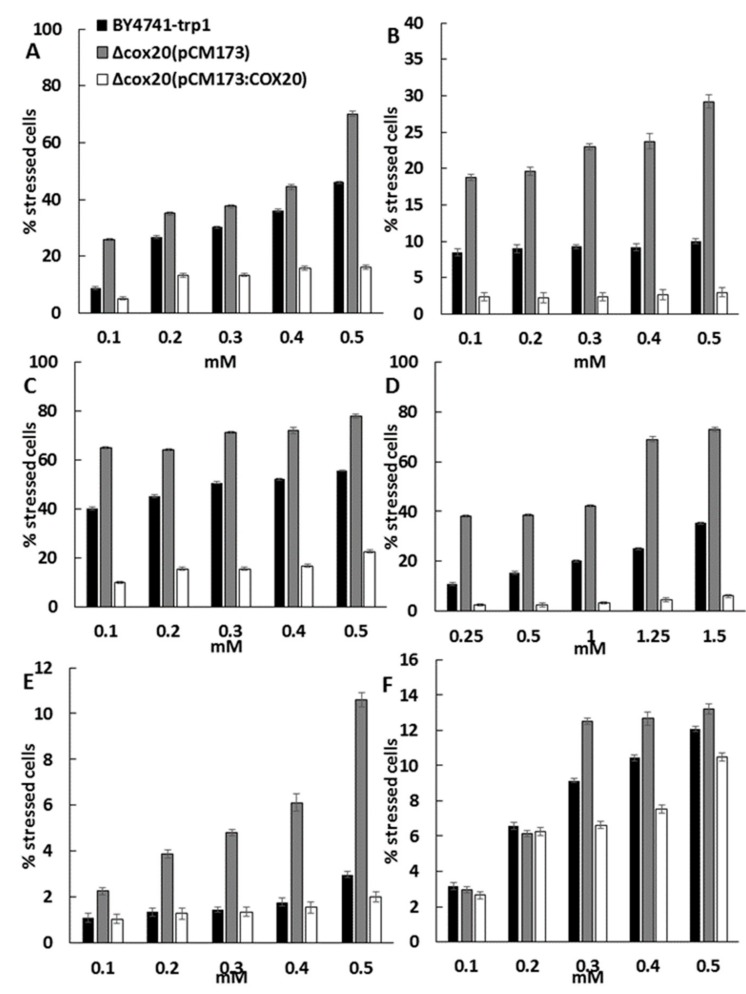
A summary of data from dot plot assays of yeast populations for BY4741-trp1, Δcox20 (pCM173), and Δcox20 (pCM173:COX20) in the presence of (**A**) 0.1–0.5 mM CuSO_4_, (**B**) 0.1–0.5 mM CoSO_4_, (**C**) 0.1–0.5 mM FeSO_4_, (**D**) 0.1–0.5 mM MnSO_4_, (**E**) 0.1–0.5 mM CdSO_4_, and (**F**) 0.1–0.5 mM ZnSO_4_. Data showing averages of triplicate assays with the standard deviation shown (*n* = 3).

**Figure 4 microorganisms-07-00575-f004:**
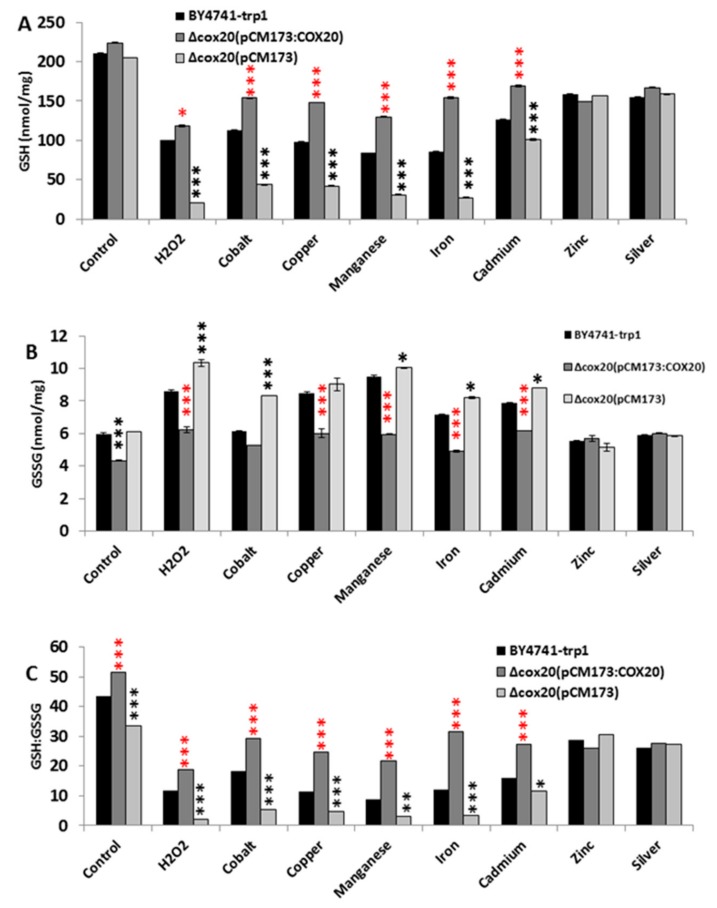
Overexpression of *COX20* reduced the presence of oxidized glutathione in BY4741-*trp1*, Δ*cox20*(pCM173), and Δ*cox20* (pCM173:*COX20*) yeast cells (**A**) reduced GSH levels in control, 1 mM hydrogen peroxide, and 0.5 mM CuSO_4_, FeSO_4_, MnSO_4_, CdSO_4_, CoSO_4_, ZnSO_4_, and AgSO_4_, respectively, (**B**) oxidized GSH (GSSG) levels in control, hydrogen peroxide, and CuSO_4_, FeSO_4_, MnSO_4_, CdSO_4_, CoSO_4_, ZnSO_4_, and AgSO_4_, respectively, (**C**) ratios of GSH:GSSG in control, hydrogen peroxide, CuSO_4_, FeSO_4_, MnSO_4_, CdSO_4_, CoSO_4_, ZnSO_4_, and AgSO_4_, respectively. Data show averages of triplicate assays with standard deviation shown (*n* = 3). Red asterisks indicate that there was a significant difference between BY4741-*trp1* and BY4741-*trp1* Δ*cox20* (pCM173:COX20), black asterisks indicate a significant difference between BY4741-*trp1* and BY4741-*trp1* Δ*cox20* (pCM173).

**Figure 5 microorganisms-07-00575-f005:**
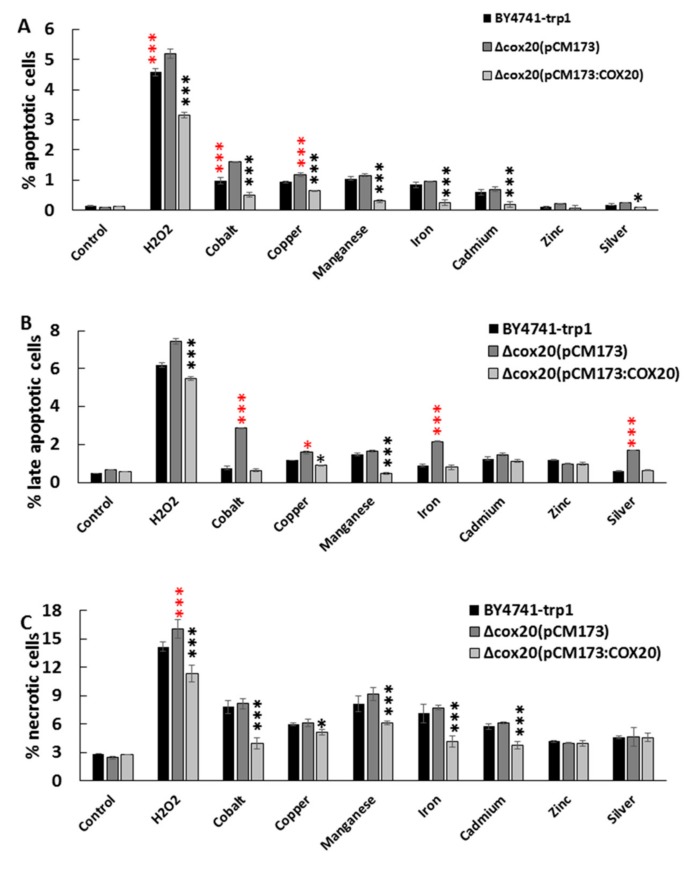
Detection of apoptosis and necrosis for BY4741-trp1, Δcox20(pCM173), and Δcox20 (pCM173:COX20) under control or oxidative stress conditions. (**A**) apoptotic cells in the presence of 1 mM hydrogen peroxide, 0.5 mM CoSO_4,_ CuSO_4_, MnSO_4_, FeSO_4_, CdSO_4_, ZnSO_4_, and AgSO_4_ (**B**) late apoptotic cells in the presence of hydrogen peroxide, CoSO_4,_ CuSO_4_, MnSO_4_, FeSO_4_, CdSO_4_, ZnSO_4_, and AgSO_4_, and (**C**) necrotic cells in the presence of hydrogen peroxide, CoSO_4,_ CuSO_4_, MnSO_4_, FeSO_4_, CdSO_4_, ZnSO_4_, and AgSO_4_ respectively. Data show averages of triplicate assays with the standard deviation shown (*n* = 3). Red asterisks indicate that there was a significant difference between BY4741-trp1 and BY4741-trp1 Δcox20 (pCM173:COX20), black asterisks indicate a significant difference between BY4741-trp1 and BY4741-trp1 Δcox20 (pCM173).

**Figure 6 microorganisms-07-00575-f006:**
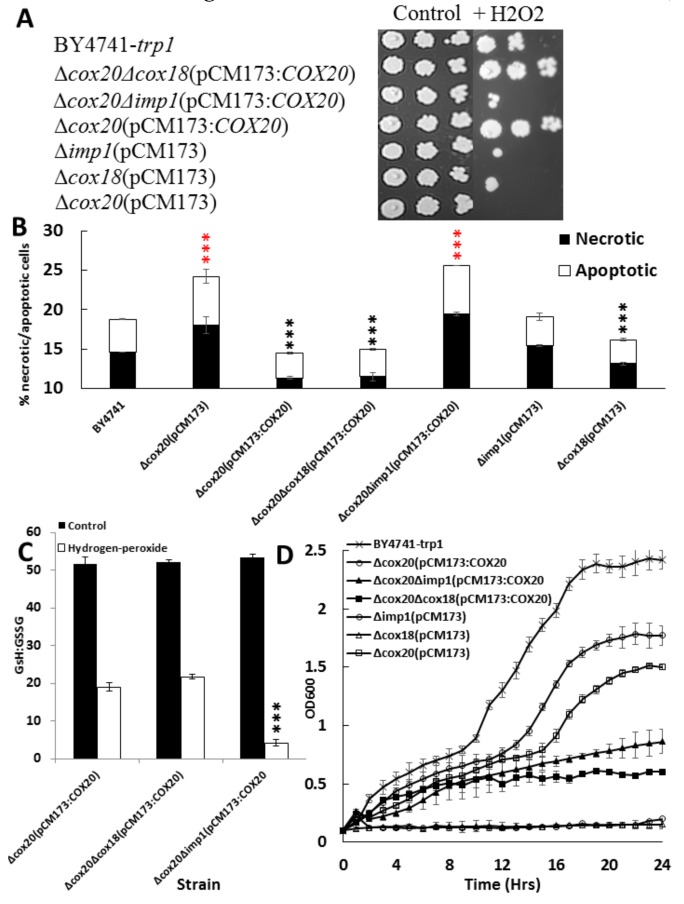
(**A**) Results of spot plate assay for yeast strains Δ*cox20* (pCM173), Δ*cox18* (pCM173), Δ*imp1* (pCM173), Δ*cox20* (pCM173:*COX20*), Δ*cox20*Δ*cox18* (pCM173:*COX20*), Δ*cox20*Δ*imp1* (pCM173:*COX20*) under control and in the presence of 1 mM hydrogen peroxide (HP) (10^7^-10^5^ cells/mL), (**B**) Percentage of apoptotic and necrotic cells of strains BY4741, Δ*cox20* (pCM173), Δ*cox18* (pCM173), Δ*imp1* (pCM173), Δ*cox20* (pCM173:*COX20*), Δ*cox20*Δ*cox18* (pCM173:*COX20*), Δ*cox20*Δ*imp1* (pCM173:*COX20*) in the presence of 1 mM hydrogen peroxide, (C) GSH:GSSG ratio for Δ*cox20* (pCM173:*COX20*), Δ*cox20*Δ*cox18* (pCM173:*COX20*) and Δ*cox20*Δ*imp1* (pCM173:*COX20*) under control and in the presence of 1 mM hydrogen peroxide, (**D**) growth on 2% glycerol for Δ*cox20* (pCM173), Δ*cox18* (pCM173), Δ*imp1* (pCM173), Δ*cox20* (pCM173:*COX20*), Δ*cox20*Δ*cox18* (pCM173:*COX20*), and Δ*cox20*Δ*imp1* (pCM173:*COX20*). Data, where appropriate, show averages of triplicate assays with the standard deviation shown (*n* = 3). Red asterisks indicate that there was a significant difference between BY4741-*trp1* and BY4741-*trp1* Δ*cox20* (pCM173:COX20), black asterisks indicate a significant difference between BY4741-*trp1* and BY4741-*trp1* Δ*cox20* (pCM173).

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
