# Peer review of "A Role for *COX20* in Tolerance to Oxidative Stress and Programmed Cell Death in *Saccharomyces cerevisiae"

_microorganisms, 2019, doi:10.3390/microorganisms7110575_

Round 1

Reviewer 1 Report

The title of manuscript    (A role for cytochrome C oxidase (COX20) in tolerance…)  needs clarification: COX20 is not a part of cytochrome oxidase complex, it is “required for proteolytic processing of Cox2p and its assembly into cytochrome c oxidase” (SGD data base)

the advantage and the interesting point in this paperis the result demonstrating that the potassium ion oscilattion in glycolysis is not due to transport across cytoplasmic and vacuolar membranes and is connected with cytosceleton functions.

Author Response

I've removed cytochrome c oxidase from the title.

I'm assuming the second sentence is associated with another manuscript?

Reviewer 2 Report

The manuscript by Keerthiraju and collaborators is a simple but useful study on the role of Cox20p in the response of Saccharomyces cerevisiae cells to oxidative stress. The authors quantified ROS levels, GSH/GSSG ratios, apoptosis/necrosis, and growth in respiratory conditions for a pool of strains deleted in various genes, including COX20 but also genes functionally related to Cox20p (IMP1 and COX18) grown in control conditions or in the presence of H2O2, pyocyanin, or metal sulphates. This way, they could confirm the central role of Cox20p in the response to oxidative stress.

Major comments:

1- the findings of the study (totally) lack of statistical support. Statistical tests have been done only for a few analyses, and I am afraid the chosen tests are not the most appropriate. The data obtained from flow cytometry have not be analysed with statistical tests, the authors simply state that in some cases there are differences… I do agree that the differences are obvious (see bar plots in figures 1 and 2) but the tests are necessary, especially considering that data are the results of only three replicates. Other parts of the manuscript highlight awkward statistical analyses/errors. For instance, why using ANOVA tests? Other minor problems, possibly typos, are reported in details below.

2- Mitochondrial are fundamental for the response to/insurgence of oxidative stress. And mitochondria may vary in number (per cell) in different growth conditions. Less mitochondria, for instance, means less cytochrome c oxidase. Hence, the number of mitochondria has a great impact on all the values the authors have quantified in their present study. However, mitochondrial quantification has been completely neglected.

3- in its current form, the manuscript fails from indicating the importance of the study. There are some references on the impact of oxidative stress on S. cerevisiae cells during fermentation, but, also considering the audience of the journal, expanding on this would be a good idea. In addition, the readers would benefit from clear statements on the importance of this study in that optic (e.g. aiming at strains more resistant to the oxidative stress and hence capable to last longer over the fermentation process).

In the discussion session, the authors emphasise the importance of acetic acid in fermentation and its (potential) role in inducing cells oxidative stress. However, the authors have not tested at all acetic acid over their experiments. Hence, either the authors tone down the emphasis on acetic acid, or they contextualise it by adding further details on the fermentation process.

Minor comments:

- Please check everywhere the way genes and proteins have been written, the standard is capital letters, Italic for genes (e.g. COX20) and plain text, only the first letter capital, and a p at the end for proteins (e.g. Cox20p). When indicating deletion, all small letters and Italic (but I think the authors correctly got this one).

-please write in the legend of every figure showing barplots what are the value reported (e.g. average, and standard deviation as error bars).

- please describe in further details the vector pCM173. Is it centromeric? Are the genes expressed from this vector under the control of a constitutive/inducible promoter?

- please add the sequences of the primers used for qRT-PCR.

- please correct the sentence at line 75.

- please divide the sentence in lines 96-102.

- pl correct apoptosis with apoptotic (L130).

- please delete the comma after “Figures”, L134.

- all the results reported from L138 to L188 need to be supported by statistical tests to corroborate the authors’ observations.

- please delete “this event was”, L138-139.

- L165-169: please complete or rephrase the sentence.

- please describe better the data reported in Figure 1K. It is barely reported that this refers to qRT-PCR data (please also note my previous comment on the description of the method). Also, clearly write that the fold change has been calculated as the value measured for COS20 divided the value measured for ACT1 (if I understood correctly).

- I believe figure 1 shown in page 6 is Figure S1? Please write on top of the scatter plot, for each column the corresponding strain (it is already reported in the figure legend, but having it written on the figure would help the reader).

- please modify figure 1 and figure S1, numbers are too small to read (axes and within the plots).

- I do not understand why the authors used two different colors for the asterisks in Figures 3, 4, and 5. If it is because otherwise asterisks referring to different comparison would overlap, they could use the same solution they used in Fig. 3B: write the asterisk in vertical.

- Related to the problems with statistics, it is in correct but hardly used by the community, to report p values as percentages as in Figure 3 and Figure 4 legends.

- it is somehow curious that the authors write about the existence of “an excellent model of how COX2 is processed” (L260) [please note here you should write Cox2p] has been constructed and was used in the presented study, but they do not describe it at all. It would be very interesting to have it briefly reported in the manuscript, especially to highlight how the model was used as a guide in the study.

- typos related with statistics: L281, L284, L296, L297, I believe the references to p values are wrong. L281 and L284 should be < (not >), L296 and L297 should be = (not >).

Author Response

Major comments:

the findings of the study (totally) lack of statistical support. Statistical tests have been done only for a few analyses, and I am afraid the chosen tests are not the most appropriate. The data obtained from flow cytometry have not be analysed with statistical tests, the authors simply state that in some cases there are differences… I do agree that the differences are obvious (see bar plots in figures 1 and 2) but the tests are necessary, especially considering that data are the results of only three replicates. Other parts of the manuscript highlight awkward statistical analyses/errors. For instance, why using ANOVA tests? Other minor problems, possibly typos, are reported in details below.

I’ve included use of coefficient variation which we used as a means of statistically measuring the flow cytometry data.

Mitochondrial are fundamental for the response to/insurgence of oxidative stress. And mitochondria may vary in number (per cell) in different growth conditions. Less mitochondria, for instance, means less cytochrome c oxidase. Hence, the number of mitochondria has a great impact on all the values the authors have quantified in their present study. However, mitochondrial quantification has been completely neglected.

I agree but the very short turnaround for this paper means I don’t have time to quantify mitochondria levels in the cells but it is something I will look into

in its current form, the manuscript fails from indicating the importance of the study. There are some references on the impact of oxidative stress on  cerevisiae cells during fermentation, but, also considering the audience of the journal, expanding on this would be a good idea. In addition, the readers would benefit from clear statements on the importance of this study in that optic (e.g. aiming at strains more resistant to the oxidative stress and hence capable to last longer over the fermentation process).

I’ve added some sentences and references to support this research

In the discussion session, the authors emphasise the importance of acetic acid in fermentation and its (potential) role in inducing cells oxidative stress. However, the authors have not tested at all acetic acid over their experiments. Hence, either the authors tone down the emphasis on acetic acid, or they contextualise it by adding further details on the fermentation process.

We looked at acetic acid during these experiments but don’t report it here as we wanted to concentrate on other chemical induced forms of oxidative stress.

Minor comments:

- Please check everywhere the way genes and proteins have been written, the standard is capital letters, Italic for genes (e.g. COX20) and plain text, only the first letter capital, and a p at the end for proteins (e.g. Cox20p). When indicating deletion, all small letters and Italic (but I think the authors correctly got this one).

I’ve changed this throughout

-please write in the legend of every figure showing barplots what are the value reported (e.g. average, and standard deviation as error bars).

I’ve added these

- please describe in further details the vector pCM173. Is it centromeric? Are the genes expressed from this vector under the control of a constitutive/inducible promoter?

pCM173 are centromeric yeast plasmids, marker TRP1, tetracycline repressed expression of lacZ. These plasmids are under the control of the tetO2 and tetO7 promoters, respectively. 

- please add the sequences of the primers used for qRT-PCR.

I’ve removed the qPCR data so don’t need to add these primers.

- please correct the sentence at line 75.

This has been corrected

- please divide the sentence in lines 96-102.

- pl correct apoptosis with apoptotic (L130).

- please delete the comma after “Figures”, L134.

I’ve changed all these

- all the results reported from L138 to L188 need to be supported by statistical tests to corroborate the authors’ observations.

I’ve added coefficient of variation data which hopefully answers this question

- please delete “this event was”, L138-139.

- L165-169: please complete or rephrase the sentence.

- please describe better the data reported in Figure 1K. It is barely reported that this refers to qRT-PCR data (please also note my previous comment on the description of the method). Also, clearly write that the fold change has been calculated as the value measured for COS20 divided the value measured for ACT1 (if I understood correctly).

I’ve decided to remove this figure

- I believe figure 1 shown in page 6 is Figure S1? Please write on top of the scatter plot, for each column the corresponding strain (it is already reported in the figure legend, but having it written on the figure would help the reader).

- please modify figure 1 and figure S1, numbers are too small to read (axes and within the plots).

I’ve modified Figures 1 and removed figure S1 replacing it with a new figure 3.

- I do not understand why the authors used two different colors for the asterisks in Figures 3, 4, and 5. If it is because otherwise asterisks referring to different comparison would overlap, they could use the same solution they used in Fig. 3B: write the asterisk in vertical.

Red asterix’s indicate there is a significant difference in response between BY4741-trp1 and BY4741-trp1 Δcox20(pCM173:COX20). Black asterixs indicate a significant difference between BY4741-trp1 and BY4741-trp1 Δcox20(pCM173). I have now put all the asterixs in vertically.

- Related to the problems with statistics, it is in correct but hardly used by the community, to report p values as percentages as in Figure 3 and Figure 4 legends.

I have removed these.

- it is somehow curious that the authors write about the existence of “an excellent model of how COX2 is processed” (L260) [please note here you should write Cox2p] has been constructed and was used in the presented study, but they do not describe it at all. It would be very interesting to have it briefly reported in the manuscript, especially to highlight how the model was used as a guide in the study.

I’ve discussed the importance of this model in greater detail.

- typos related with statistics: L281, L284, L296, L297, I believe the references to p values are wrong. L281 and L284 should be < (not >), L296 and L297 should be = (not >).These have all been altered.

Reviewer 3 Report

The manuscript “A role for cytochrome C oxidase (COX20) in tolerance to oxidative stress and programmed cell death in Saccharomyces cerevisiae” by Keerthiraju E et al. presents data on the role of COX20 in tolerance of S. cerevisiae to oxidative stress induced by hydrogen peroxide and some metal ions. Despite some findings are interesting, the manuscript is quite superficial. It does not reveal the molecular mechanism of how COX2 protects the yeast cells against oxidative insults. In my opinion, the authors should validate more markers of oxidative stress and cell death. What is also missing and makes the manuscript data shy is lack of concentration-dependent analysis of tested yeast mutants against oxidative insults. Only a single concentration was tested (1 mM H2O2 or 0.5 mM metal ions) and discussed in the manuscript.

Additional comments:

Lane 63 – How was the deletion of COX18 and IMP1 verified? By PCR or sequencing? Please, specify.

Lanes 71-72 – What was the solution used to resuspend the cells before adding glass beads? The sentence “Cells were broken with glass beads using a MagNalyser bead beater for 30 seconds at 4°C, before incubation on ice for 15 min after breaking the cells to precipitate the proteins“ is misleading. The extraction buffer has to be specified. Proteins do not simply precipitate on ice.

Lane 85 and Lane 103 – Why there is a difference in cell density measured at OD600 for spot plate assay – OD600 of 1.0/107 cells/ml vs. determination of GSH – OD600 of 1.0/2.5x107 cells/ml?

Lanes 107-113 – Repeating sentences, please rephrase.

Lanes 130-131 – The same as above, reformulate the last sentence of this paragraph.

Lanes 138-144 – It is not necessary to explain how to analyse the flow cytometry data. Please, delete the whole paragraph and continue with lane 145.

Lane 150 – Figure 1 – Please, provide better resolution of flow cytometry charts. It seems that gating is not the same for all samples despite the authors declared that the same quadrant gates were used for subsequent analysis of mutant cells (lane 142). If it´s the case, correct the values of Figure 1 and lanes 160-169.

Lane 195 – Please, provide better resolution of Figures S1 and correct the legend to Figure S1.

Lane 187-188 – Did authors try to test also higher (above 0.5 mM) concentrations of cadmium, silver and zinc ions? If so, please include comments in the manuscript.

Lane 267 – Please, provide new Figure5A where all the strains/mutants were tested together, mean on one plate for control/untreated and one plate supplemented with hydrogen-peroxide. It is unacceptable to cut-out and put together data of spot plate assays from various plates. Especially, when testing the sensitivity of various mutants to DNA damaging agents! Also, please unite the label for hydrogen peroxide (preferentially use H2O2).

Author Response

Additional comments:

Lane 63 – How was the deletion of COX18 and IMP1 verified? By PCR or sequencing? Please, specify.

The deletion of both these genes was confirmed by sequencing

Lanes 71-72 – What was the solution used to resuspend the cells before adding glass beads? The sentence “Cells were broken with glass beads using a MagNalyser bead beater for 30 seconds at 4°C, before incubation on ice for 15 min after breaking the cells to precipitate the proteins“ is misleading. The extraction buffer has to be specified. Proteins do not simply precipitate on ice.

I’ve decided to remove the qPCR data so this information isn’t required in the manuscript

Lane 85 and Lane 103 – Why there is a difference in cell density measured at OD600 for spot plate assay – OD600 of 1.0/107 cells/ml vs. determination of GSH – OD600 of 1.0/2.5x107 cells/ml?

I’ve altered this so it reads 1.0 x 107 for both

Lanes 107-113 – Repeating sentences, please rephrase.

Lanes 130-131 – The same as above, reformulate the last sentence of this paragraph.

These have been changed

Lanes 138-144 – It is not necessary to explain how to analyse the flow cytometry data. Please, delete the whole paragraph and continue with lane 145.

I’ve deleted this paragraph

Lane 150 – Figure 1 – Please, provide better resolution of flow cytometry charts. It seems that gating is not the same for all samples despite the authors declared that the same quadrant gates were used for subsequent analysis of mutant cells (lane 142). If it´s the case, correct the values of Figure 1 and lanes 160-169.

I’ve provided a larger figure which shows the same quadrant gates were used throughout

Lane 195 – Please, provide better resolution of Figures S1 and correct the legend to Figure S1.

I've decided to replace S1 with a much expanded figure 3.

Lane 187-188 – Did authors try to test also higher (above 0.5 mM) concentrations of cadmium, silver and zinc ions? If so, please include comments in the manuscript.

This data has now been included in the new version of figure 3

Lane 267 – Please, provide new Figure5A where all the strains/mutants were tested together, mean on one plate for control/untreated and one plate supplemented with hydrogen-peroxide. It is unacceptable to cut-out and put together data of spot plate assays from various plates. Especially, when testing the sensitivity of various mutants to DNA damaging agents! Also, please unite the label for hydrogen peroxide (preferentially use H2O2).

I've put in a new figure 5A and changed Hydrogewn peroxide to H2O2

Round 2

Reviewer 3 Report

-